# Aquatic Freshwater Vertebrate Models of Epilepsy Pathology: Past Discoveries and Future Directions for Therapeutic Discovery

**DOI:** 10.3390/ijms23158608

**Published:** 2022-08-03

**Authors:** Rachel E. Williams, Karen Mruk

**Affiliations:** School of Pharmacy, College of Health Sciences, University of Wyoming, Laramie, WY 82071, USA; rwilli54@uwyo.edu

**Keywords:** drug development, zebrafish, *Xenopus laevis*, seizures, drug screening

## Abstract

Epilepsy is an international public health concern that greatly affects patients’ health and lifestyle. About 30% of patients do not respond to available therapies, making new research models important for further drug discovery. Aquatic vertebrates present a promising avenue for improved seizure drug screening and discovery. Zebrafish (*Danio rerio*) and African clawed frogs *(Xenopus laevis* and *tropicalis)* are increasing in popularity for seizure research due to their cost-effective housing and rearing, similar genome to humans, ease of genetic manipulation, and simplicity of drug dosing. These organisms have demonstrated utility in a variety of seizure-induction models including chemical and genetic methods. Past studies with these methods have produced promising data and generated questions for further applications of these models to promote discovery of drug-resistant seizure pathology and lead to effective treatments for these patients.

## 1. Introduction

Epilepsy is a current public health concern with more than 50 million people affected around the world [1]. This disease takes a toll on the health and lifestyle of affected patients in the form of discrimination, comorbidities, and the constant stress of managing a chronic disease [2]. About 70% of patients attain adequate seizure control with available medications, but up to one-third of patients do not respond to available treatments [2,3,4,5,6,7]. The search for new epilepsy treatments requires biological models that can mirror the complexity of the human nervous system and allow researchers to discover new drug targets and drug molecules. Aquatic vertebrates provide promising alternative models for seizure modeling and drug screening and development. Zebrafish (*Danio rerio*) and *Xenopus* species of frog are two emerging aquatic models for seizure physiology and drug screening.

### Classification and Etiology of Seizures

Seizures are bursts of excessive electrical activity in the brain. Uncontrolled electrical activity can lead to temporary abnormalities in both behavior and states of awareness. The International League Against Epilepsy (ILAE) has developed a standardized classification system for seizures [8] (Figure 1). Clinicians use guidelines for first-line treatment based on the seizure classification [9,10]. Overall, anti-seizure therapeutics vary based on the molecule’s pharmacokinetic properties, potential drug–drug interactions, side effects, and toxicities; however, most work mechanistically to suppress seizures [11]. Despite the large number of anti-seizure compounds on the market, drug-resistant epilepsy occurs in up to 30% of patients, refs. [2,3,4,5,6,7] and initial treatment fails in an additional 30% of patients due to intolerable side effects [12]. Discovery of therapeutic compounds that could affect disease progression is a required new direction for anti-seizure therapeutic discovery [13].

An additional challenge in developing effective therapies for seizures is that there are many underlying causes for seizures. For example, most seizures in newborns are often classified as acute symptomatic seizures and are usually due to a brain injury [14]. In contrast, neonatal-onset epilepsies are reported in ~12% of newborns with seizures and can be associated with genetic mutations [15,16,17,18,19], brain malformations [20,21], and metabolic disorders [22,23]. Model organisms that are amendable to genetic manipulations, brain imaging, and high-throughput screening of new compounds are critical for discovery of new therapeutics with different properties.

## 2. Aquatic Freshwater Vertebrate Animal Model Advantages

Both zebrafish (*Danio rerio*) and frogs (*Xenopus laevis* and *Xenopus tropicalis*) are cornerstones in biomedical research. They offer many advantages over the traditionally used mammalian models. Both zebrafish and frogs can be more easily raised and housed in large numbers, particularly at larval stages [24,25]. The transparency of zebrafish embryos and *Xenopus* tadpoles permits live imaging at the organismal level with a number of tissue-specific transgenic lines available to permit tracking of cellular dynamics in vivo [26,27]. High-throughput screens are possible for both drug discovery and toxicology [28,29,30,31,32,33,34]. A number of compounds identified in zebrafish screens are in early clinical trials further underscoring the translational potential of conducting drug screens in zebrafish [35]. With the advent of whole-genome sequencing, both zebrafish and *Xenopus* have stood out as model organisms among their mammalian counterparts. Over 70% of human genes have at least one zebrafish orthologue [36], with the *Xenopus* genome including orthologs of ~80% of human disease genes [37]. More recently, methods to increase the genome editing efficiency of the Clustered Regularly Interspaced Short Palindromic Repeat– (CRISPR–) Cas9 system in zebrafish [38,39,40,41,42,43] and *Xenopus* [44,45,46,47,48] have led to new human disease models.

Although zebrafish and *Xenopus* do not have the same level of regulatory recognition as their mammalian counterparts, government agencies such as the National Toxicology Program are now funding programs to increase the utility of aquatic vertebrates as pre-clinical models [49]. Indeed, a number of academic and pharmaceutical companies have teamed up with contract research service (CRO) companies that focus on zebrafish and *Xenopus* for preclinical drug development [50]. A growing list of drug treatments that have recently entered clinical trials after research starting in zebrafish is reviewed in [35].

### 2.1. Aquatic Freshwater Vertebrates as Seizure Models

*Xenopus laevis* oocytes have a long history of contributing to epilepsy research. Intact oocytes are a versatile expression system for functional investigation of ion channels and transporters [51]. As early as the 1980s, laboratories isolated RNA from mammalian brains, injected total RNA into oocytes, and recorded membrane currents [52,53,54,55,56,57]. Later, laboratories cloned and injected ion channel mRNA to characterize how genetic mutations linked to epilepsy affect the biophysical properties of specific channels [58,59]. Furthermore, treatment with epileptogenic agents identified the role voltage-gated potassium (Kv) channels and N-methyl-D-aspartate (NMDA) receptors play in seizure generation [60,61].

At the organismal level, zebrafish larvae have been used to model seizure disorders for some time while *Xenopus* tadpoles are a more recently developed aquatic vertebrate gaining notoriety in the field. Both of these organisms have advantages as models for seizures versus traditional mammalian models. In addition to the efficient rearing, low housing costs, and transparency described above, the central nervous system (CNS) of the zebrafish and *Xenopus* have similar organization to the human CNS [62,63,64,65,66,67,68,69]. Zebrafish and *Xenopus* have myelinated axons and are used as models for demyelination [70,71,72,73,74,75]. Aquatic freshwater vertebrate models of demyelination are particularly useful given that children with epilepsy have abnormal myelin development [76,77] and patients with demyelinating disease also suffer from seizures [78]. Lastly, zebrafish and *Xenopus* use gamma-aminobutyric acid (GABA) and glutamate receptors to control CNS activity including signaling required for movement [79,80,81,82,83]. Therefore, the process of seizure induction between freshwater aquatic vertebrates and humans are likely conserved, as GABA and glutamate are the main neurotransmitters contributing to the pathophysiology of epilepsy.

### 2.2. Advantages of Aquatic Freshwater Vertebrates as Seizure Models

A major advantage of zebrafish larva and *Xenopus* tadpoles is the ability to visualize the nervous system during seizure activity in real time. Using calcium imaging, scientists can look at both excitatory and inhibitory activity throughout the brain, identify areas of high activity, and assess the rate of spread of the seizure to other parts of the brain [84,85,86,87,88,89,90,91]. Calcium imaging can be combined with a variety of electrophysiological approaches to get a direct correlation between neuronal activity and electrical field potential [92,93,94,95]. With recent advantages in calcium imaging, it is now also possible to get single cell resolution in an acute zebrafish seizure model [96].

In addition to their transparency, the aquatic nature of these organisms allows for easy drug delivery by dosing the water they reside in. Specific concentrations of drugs can be delivered by creating various baths and placing the desired animals in the solution. Drug exposure using this method allows for control of dose, frequency, and length of exposure more so than administering a drug orally or parenterally. The ability to cultivate large numbers of larvae or tadpoles allows for high-throughput screening options for drugs [97]. By dosing the water and allowing zebrafish and *Xenopus* to behave freely, the data are not confounded by anesthetics or invasive surgical procedures as found in mammalian models providing cleaner results due to freedom from interference in the neural circuits [98]. Taken together, aquatic vertebrates provide an animal platform with improved control of dose- and timing-related phenomena.

Perhaps the largest advantage aquatic freshwater vertebrates have in modeling seizures is their genetic tractability. Recent studies have found approximately 900 genes associated with epilepsy [99], making genetic animal models of seizures ever more useful for studying the disease. The ease of genetic manipulation in zebrafish and *Xenopus* affords them a major advantage over mammalian models as mutations in multiple types of proteins can be made in combination. For example, genetic seizure disorders are highly variable and prone to drug resistance [100]. However, it is less understood whether resistance is due to mutations that cause changes in the pharmacokinetics and pharmacodynamics of the therapeutic molecules or inherent differences in the pathophysiology of the seizure. The extensive genetic toolkit available in aquatic freshwater vertebrates permits modeling of knockdown [101,102,103,104,105,106,107], knockout [39,44,46,48,108,109,110,111,112,113,114,115,116], ectopic, and overexpression [38,117,118,119,120,121] of seizure-associated genes in combination with metabolic enzymes, transporters, and other proteins required for absorption, distribution, degradation, and excretion of therapeutics. A unique aspect of *Xenopus* is that their large eggs and embryos can be genetically modified by injecting the embryo on only one side at the two-cell stage, providing an internal control [120]. This causes the alteration to occur only on the injected side, so the resulting phenotype can be compared to the contralateral side for off-target effects. Seizure activity can be monitored by similar behaviors in both species in combination with calcium imaging and electrophysiology.

Lastly, researchers in both model organism communities use a similar rating scale for seizures. The rating scale consists of five categories ranging from barely noticeable locomotor changes (category 1) to C-shaped contractions (category 5) [98,122,123,124]. Although zebrafish can exhibit some human symptoms, such as tonic-like behavior, this scoring system does not directly relate to the ILAE classification. Instead, this scoring system provides a method for standardization of experimental technique, which allows the data to be reproducible and translatable across laboratories and different aquatic species. Other behavioral metrics used to analyze seizure activity include total distance traveled and thigmotaxis.

### 2.3. Disadvantages of Aquatic Freshwater Vertebrates as Seizure Models

Like most model organisms, both zebrafish and *Xenopus* have disadvantages. Adults develop pigment, making live imaging more difficult in older animals. Genetic mutants which lack pigment may alleviate live imaging limitations of adult animals. As techniques are developed to image brain activity in older animals, the use of juvenile and adult models of seizure disorders should increase. Indeed, a genetic model of juvenile myoclonic epilepsy in older zebrafish has demonstrated convulsive seizure generation in response to light [125]. More recently, an EEG system was developed for adult zebrafish, permitting a direct comparison between zebrafish and mammalian models [126].

Although dosing the water is facile, uptake of the compounds through skin and gills varies, creating pharmacokinetic challenges such as indirect measurements of drug absorption into plasma. Though most studies in zebrafish do not measure blood concentration of compounds, these measurements could prove essential for translating the therapeutic potential of aquatic vertebrate models to the clinic [127]. Given blood samples from both zebrafish and *Xenopus* are small, pooled samples may be necessary to get the required pharmacokinetic data [128,129]. Small molecules with poor water solubility are also not absorbed efficiently, limiting the chemical libraries that can be screened by adding the molecule to the water. Furthermore, drugs which do not readily cross the mammalian blood–brain barrier are also not observed in the zebrafish brain after water administration [130].

## 3. Current Aquatic Vertebrate Seizure Models

Chemical and genetic seizure induction are the most common options in the zebrafish and *Xenopus* communities. Below we highlight the progress made toward understanding seizure pathophysiology and treatment using these methods.

### 3.1. Chemical Induction of Seizures

Seizures in humans are due to hyperexcitability and hypersynchronous activity of cortical neurons. Therefore, most chemically induced models of aquatic vertebrates are generated using compounds that disrupt the inhibitory and excitatory balance in the brain of the animal (Figure 2). Possible chemical induction agents include bicuculline, picrotoxin, tetramethylenedisulfotetramine (TETS), kainic acid, pilocarpine, 4-aminopyridine (4-AP), and pentylenetetrazole (PTZ) [98]. Most of the studies in aquatic organisms that have utilized non-PTZ chemical induction methods are in zebrafish. Chemical induction agents are advantageous due to rapid seizure induction and ease of use. They also facilitate high-throughput screening by permitting rapid dose response studies in multiple animal ages to determine desired characteristics prior to the main data-generating experiment. Zebrafish studies have used animals as young as 2 days post-fertilization (2 dpf) up to adulthood whereas the *Xenopus* community typically uses tadpoles at developmental stages 42–49.

Bicuculline is a competitive GABA_A_ antagonist [131] originally used to elucidate details surrounding synapses and GABA_A_ transmission [132,133]. It is a light-sensitive molecule, which easily decomposes in solution, making it difficult to administer [131]. Therefore, studies using this compound are limited in free-swimming aquatic vertebrates. One advantage of bicuculline is the ability to induce seizures in multiple models from *C. elegans* to cats [134,135,136], providing a clear translational path from target identification in aquatic organisms to mammals. In rats, natural product screens identified sakuranetin and melittin as protective against bicuculline-induced seizures [137,138]. However, exposure to bicuculline does not cause significantly different electrical recordings than those generated from other chemically induced seizures [134]. Given the difficulty working with bicuculline and lack of unique electrical changes induced, its utility in aquatic vertebrates is still limited.

Picrotoxin is also a GABA_A_ antagonist but unlike bicuculline, is a non-competitive inhibitor [132,133]. In zebrafish larvae (5 dpf), picrotoxin exposure increased locomotion in a dose-dependent manner. Similarly, *Xenopus* tadpoles also displayed concentration-dependent seizures [98]. In addition, high doses increased thigmotaxis in zebrafish larvae, which is a common measure for anxiety. Furthermore, higher doses of acute anti-seizure medications are required to decrease locomotor seizure symptoms when induced with this compound [94]. Therefore, picrotoxin may be advantageous in modeling treatment-resistant seizures. Picrotoxin-treated adult zebrafish exhibited increased hyperactivity and cortisol levels following seizure [139]. The data from larvae and adults are consistent with studies where low doses of picrotoxin elevate anxiety and corticosterone in mice [140] and lysosomal dysfunction in rats [141,142]. Therefore, the picrotoxin model in adult zebrafish may provide a useful tool to probe seizure-induced effects on the endocrine system. The seizure-inducing properties of picrotoxin makes it a credible chemical threat to humans. In fact, the NIH Countermeasures Against Chemical Threats (CounterACT) program listed picrotoxin as an agent of interest. In addition to seizure modeling, picrotoxin in zebrafish is a good platform to screen symptoms and potential treatments for the use of picrotoxin against human populations in warfare [94].

Similar to picrotoxin, TETS is also a non-competitive GABA_A_ antagonist that is considered a potential chemical warfare agent. While TETS is a potent rodenticide, it also causes human seizures with lasting neurological effects, leading to a worldwide ban [143]. The recurrent nature of seizures after TETS exposure in humans as well as the lack of a targeted treatment make it attractive as a seizure-induction agent for research using aquatic vertebrate models. Compared to picrotoxin, in zebrafish larvae less TETS is required to evoke seizure behavior [94,144]. TETS also triggered high-frequency electrical discharges which were different from electrical measurements following picrotoxin exposure. Benzodiazepine treatment in larvae attenuated some of the electrical changes without a full return to baseline, consistent with what is seen in human exposure. Future studies aimed at identifying TETS-protective compounds in zebrafish will facilitate the development of effective antidotes for this poison.

Kainic acid is an analog of glutamate, which promotes excitability leading to seizure activity. It activates both kainate and AMPA receptors. One advantage of this drug is that it causes focal seizures in both mammals and primates [145,146,147]. Kainic acid causes cell death and damage to the brain, but zebrafish can regenerate brain cells [148]. Upon regeneration, after kainic acid treatment, the brain becomes disorganized which leads to a chronic spontaneous seizure state similar to epilepsy in humans. One study showed a lack of response of this seizure model to clinically used anti-seizure medications [149]. This lack of response underscores the potential of this model to identify drugs for treatment of drug-resistant seizures. The major disadvantage of this model is that even in aquatic organisms, kainic acid is not well absorbed and must be injected, making administration more technically challenging and time intensive.

Pilocarpine induces seizures through activation of the cholinergic system in the brain, specifically agonizing the muscarinic (M1) receptor. Activation of the cholinergic system leads to activation of NMDA receptors and hyperexcitability [150]. Some evidence suggests that pilocarpine seizures have different behavioral characteristics than PTZ-induced seizures. This makes pilocarpine beneficial for modeling different types of seizures possibly leading to new therapeutic options [150]. Multiple studies have used pilocarpine especially to model seizures in adult zebrafish with emphasis on chronic seizure modeling through repeated dosing [150,151]. Repeated dosing is particularly attractive because patients typically present with recurrent seizures, so modeling this will allow for better drug discovery leading to treatment of recurrent versus acute seizures.

4-AP is a voltage-sensitive potassium (K+) channel blocker. K+ channel inhibition by 4-AP results in increased cholinergic signaling in the CNS and neuromuscular junctions resulting in clonic seizures [152]. 4-AP was originally used to repel and kill birds, and it is toxic to mammals as well [153]. Despite its toxicity, 4-AP is used as a therapeutic for multiple sclerosis and spinal cord injury patients [154,155,156]. Therefore, 4-AP-induced seizures are a good model for seizure-threshold lowering therapeutics. In *Xenopus*, 4-AP induces both behavioral and rhythmic high-amplitude electrical discharges [98]. Similarly, zebrafish larvae also exhibit increased swimming activity when treated with 4-AP. Recently, a screen to test the efficacy of current anti-seizure compounds was performed in hippocampal-entorhinal slices of adult rats with 4-AP-induced seizures [152]. Given the ease of drug application in aquatic vertebrates, it is possible to simultaneously administer a seizure-threshold lowering drug such as 4-AP at a low dose and screen for seizure-protective compounds.

The most common chemical induction model in zebrafish and *Xenopus* is PTZ. PTZ is a GABA_A_ antagonist and as such PTZ-induced seizures are sensitive to drugs acting directly on GABA_A_ receptors. The PTZ model is ideal for its translation to mammalian models. PTZ has been used to induce seizures in rodents [157], canines [158], and primates [159]. Using the PTZ model, early zebrafish studies focused on zebrafish larvae and their changes in electrical activity [97]. Pharmacologically, PTZ seizures and identified anti-seizure compounds from these screens in zebrafish larvae are consistent with results from rodent studies [160,161]. The PTZ-induced seizure model in zebrafish larvae has also been used to screen a variety of natural compounds for anti-seizure activity [162,163,164,165]. These screens aimed to discover novel anti-seizure therapies for treating refractory seizure disorders. While larvae may be more versatile for imaging and replicating developmental neural tissue, adult zebrafish still have a role in anti-seizure drug research with potential applications to older human patients. One study used adult zebrafish instead of larvae to screen the anti-seizure properties of leaf extracts [166]. Taken together, the PTZ-zebrafish model is a first-line drug-screening tool for discovery of anti-seizure medications. Further, the model shows promise as a first line tool for identifying the adverse effects of seizure induction in medications indicated for other diseases.

*Xenopus* models have also demonstrated efficacy as a drug-discovery tool for seizures. Most *Xenopus* studies use PTZ for chemical seizure induction. In fact, the Haas laboratory tested the different seizure-induction agents described above to determine the most consistent seizure model in *Xenopus* [98]. PTZ is preferred for its wide therapeutic window and consistent timing and intensity of seizure activity in the tadpole [98]. Similar to zebrafish, the *Xenopus* community uses intracellular calcium variations to complement locomotor assays. Furthermore, the PTZ-induced *Xenopus* seizure model exhibits electrical events consistent with those seen in other vertebrates [167].

### 3.2. Genetic Induction of Seizures

Originally, genetic seizure models were generally limited to forward-genetic screens using N-ethyl-N-nitrosourea (ENU) mutagenesis. As genetic engineering techniques bloomed, multiple models became available. For example, gene knockdown with antisense morpholinos has been used to model autosomal dominant partial epilepsy with auditory features and temporal lobe epilepsy [168,169]. Morpholino knockdown can also be used to probe the mechanism of disease-linked genes in epilepsy formation [170]. Unfortunately, morpholinos come with their own set of limitations including off-target effects, transient effects that wane after days, and noted discrepancies between morphants and mutant phenotypes.

Given the growing list of concerns surrounding morpholinos, laboratories derived genetic seizure models from clinically relevant mutations in humans that resulted in seizure disorders. The Baraban laboratory published a phenotypic analysis of 40 single-gene mutant zebrafish lines based on genes implicated in childhood epilepsy [171]. For example, Dravet syndrome is due to mutations in the Na_V_1.1 voltage-gated sodium channel. A phenotype-based screen in mutant zebrafish larvae identified clemizole as an anti-seizure compound [172]. In addition, zebrafish with mutations in the syntaxin binding protein, *stxbp1*, also respond to clemizole [173]. These and future genetic models are expected to provide much needed information on the pathophysiology of childhood epilepsies and identify new classes of anti-seizure compounds paving the way for patient-specific therapeutics.

*Xenopus* models are also genetically amendable. Using cRNA injection technology, a *Xenopus* oocyte model was used to find a non-conventional seizure treatment with direct application to a human patient [174]. This study directly demonstrates the translational relevance of data from these aquatic vertebrates. A *Xenopus* model using CRISPR-Cas9-mediated genome editing to deplete *neurod2*, a gene implicated in early infantile epileptic encephalopathy, induces spontaneous seizures in tadpoles mimicking the human disease [175]. In addition to genetic manipulation, *Xenopus* tadpoles and oocytes have been used to determine molecular details of clinically utilized and novel drug molecules [176,177,178]. For example, the commonly used drug for absence seizures, ethosuximide, inhibits specific G-protein activated inwardly rectifying potassium channels (GIRK) which has the potential to affect other systems beyond the brain [179]. Another study demonstrated how multiple compounds can interact pharmacodynamically to produce increased effects for seizure treatment [180]. Additionally, the effects of polyamine synthesis during seizure activity are controversial. Data from the *Xenopus* model support the hypothesis that polyamines are protective against recurrent seizure activity which may indicate novel therapeutic targets [181]. As there are data indicating polyamines are not protective in mammals [182], further studies across species may be necessary to make a final determination as to the protective effect of post-seizure polyamine generation.

### 3.3. New but Less Established Aquatic Vertebrate Seizure Models

Not all anti-seizure drugs are active against multiple types of induced seizures. Screening of currently approved anti-seizure drugs showed differences in efficacy against PTZ-induced seizures and one model of genetic seizures [183]. To identify new models for anti-seizure efficacy, the Kurrasch laboratory developed zebrafish monitoring for seizure activity using mitochondrial respiration [183]. This platform identified a new compound, vorinostatcan that decreased seizures in both genetic and chemical induction models of epilepsy. Unfortunately, locomotor seizure activity does not correlate well with increased mitochondrial respiration, requiring more refinement of this model.

Zebrafish have also been used to model traumatic brain injuries and the subsequent seizure pathology seen in humans [184]. Pre-treatment of injured zebrafish with sonic hedgehog signaling inhibitors after injury rendered them resistant to anti-seizure compounds. This combinatory model of injury and drug resistance has potential as a model for drug-resistant epilepsies as these seizures are characteristically difficult to control. Traumatic brain injury models may prove useful for understanding other adult-onset acquired seizure disorders. For example, one study evaluated the development of seizures in zebrafish after injury and then tested the effectiveness of anti-seizure therapeutics using a T-maze setup to test cognitive ability before and after treatment [122]. This study highlights the importance of using multiple types of data (electrophysiological and locomotion) to track seizure activity and the effectiveness of potential treatments. Given the rise of another aquatic vertebrate, Medaka (*Oryzias latipes*), for studying traumatic brain injuries [185], this paradigm has the potential to expand our ability to model acquired seizure disorders in both a pro-regenerative and less regenerative species.

## 4. Looking to the Future for Aquatic Seizure Models

Although the PTZ model remains the most characterized and commonly used chemically induced seizure model from zebrafish to primates, aquatic vertebrates provide an opportunity to develop and validate new chemically induced seizure models. Using a variety of compounds that act through distinct pathways to induce seizures provides a broader set of models for screening anti-seizure therapeutics. These additional chemical models may uncover different anti-seizure therapeutics that could benefit treatment-resistant epilepsies in humans. Similarly, studies have shown that not all clinically effective antiepileptic drugs are active in a single seizure-induction model; therefore, aquatic vertebrates provide a model in which to test potential new drug compounds against multiple induction models before declaring them not useful. This leads to the possibility that a new pro-convulsive agent, that has yet to be considered, could be the key to modeling treatment-resistant seizure disorders.

In addition to chemically induced seizure models, aquatic vertebrates offer numerous genetic options. Stable genetic models are available for specific epilepsies such as Dravet syndrome and epilepsies linked to rare syndromes such as Angelman syndrome. Many of these models serve as a starting point for future studies aimed at gaining mechanistic insight into seizure generation and propagation. Furthermore, screening for anti-seizure therapeutics in these models permits the discovery of gene-specific therapeutics opening the door for personalized medicine in human patients.

Aquatic vertebrates permit the combination of different seizure models to determine the differences in seizure pathology and thus provide a path toward novel therapeutic options especially for drug-resistant epilepsies. The ease of housing, large clutches, transparent nature, aquatic environment, and potential for high-throughput screening make zebrafish and *Xenopus* ideal model organisms for research into the pathology and treatment of seizure disorders. The plethora of genetic resources in these models promises to provide new research opportunities and drug development directions. As genetic models are developed in new aquatic vertebrates, such as Medaka, the ability to compare evolutionary development of seizures and mechanisms of drug-resistance across models exists [186].

Continued development of imaging and screening approaches in aquatic vertebrates will increase the number and types of neurophysiological questions that can be addressed when studying seizure generation and treatment. The combination of new tools and genetic resources will accelerate the contribution of studies to translational research.

## Figures and Tables

**Figure 1 ijms-23-08608-f001:**
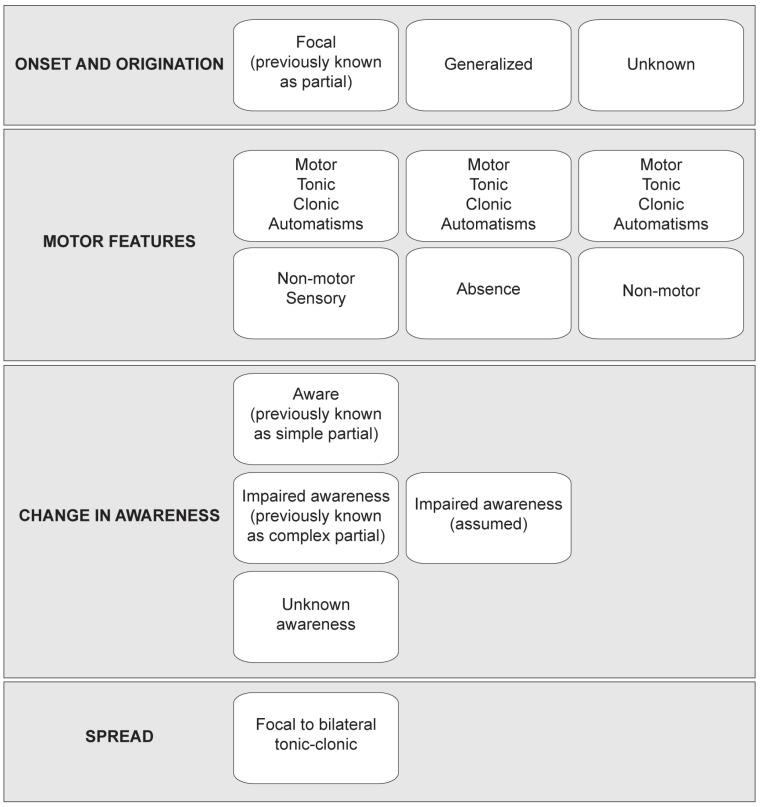
Classification of seizures according to the International League Against Epilepsy.

**Figure 2 ijms-23-08608-f002:**
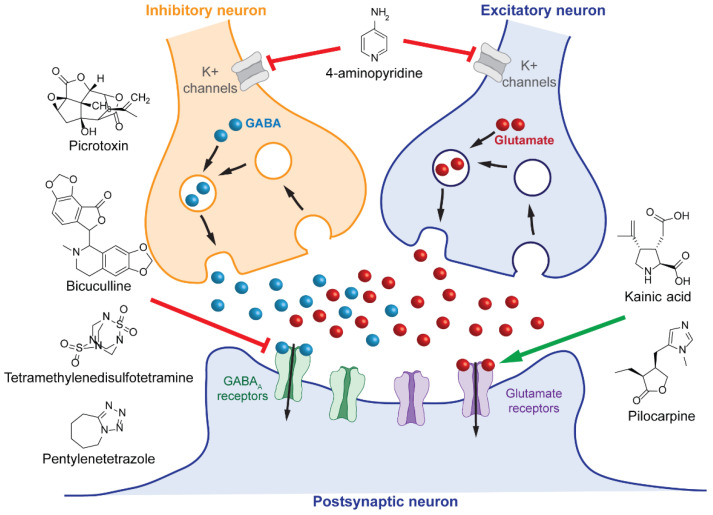
Simplified schematic of chemical inducing agents used to model seizures in aquatic vertebrates.

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
