# Peer review of "Aquatic Freshwater Vertebrate Models of Epilepsy Pathology: Past Discoveries and Future Directions for Therapeutic Discovery"

_ijms, 2022, doi:10.3390/ijms23158608_

Round 1

Reviewer 1 Report

Dear authors, In your review paper, you show how zebrafish and clawed frogs can be used in epilepsy research. The topic is interesting and the ms very well written. The manuscript is structured correctly.  The amphibian Xenopus laevis is not the most used experimental animal in the research laboratory, this species has become an important experimental model for several lines of research. 

There are some minor points that that should either be added or commented.

Abstract: Add "clawed frog" (Xenopus) or even African clawed frog (Xenopus laevis)

Keyword:  you could add "seizures" , "drug screening"..
Introduction:  Is it known that Xenopus Levis or zebrafish can also be used for regulatory testing of drug candidates? Or can the species only be used for basic research and screening? If possible, the topic should be addressed in the introduction or in the outlook. Sometimes it is unclear whether larvae are used for imaging or screening experiments or also adults, this could be worked out better.

Figure 1 corresponds in large parts to the figure from the cited paper. However, terms such as spasm and eyelid myoclonus are no longer listed. Are there reasons for this? Otherwise, all aspects of the original work should be taken up. I would suggest to show the symptoms observed in frogs and zebrafish in a table and to link the clinical pictures in human medicine e.g. Dravet Syndrome, Nav1.1. with examples from the literature. With this overview it should be possible for the reader to search specifically for models that offer alternatives to mammalian models.  

line 53: Mammals (e.g. mice) can also be kept in large numbers. The statement should be weakened. Rather, larvae and other developmental stages can of course be housed in a smaller area. For adult frogs, on the other hand, there are also limits to stocking densities. 

line 67:Can it be stated since when clawed frogs have been used in epilepsy research. Reference?

line 87-102Are there also limitations in the methods and clear advantages in the implementation for aquatic species- The individual aspects (advantages and disadvantages in dosing, sampling) should be clearly presented separately. 

line 120-125. Here you could make an illustration from the scores - based on Figure 1 and the types of epilepsy in human medicine.

Figure2:  Pre- and postsynapse could still be labelled. Also the colour of the neurotransmitters (excitatory/inhibitory). A small legend to the illustration can help understanding.

I suggest that a table be created from the molecules and epilepsy models mentioned in the text showing which molecules are used in zebrafish and/or clawed frogs, which form of epilepsy can be triggered, etc. with reference to the key publications. This would provide an easily accessible overview for comparability also with the mammalian models

Author Response

Reviewer 1

Concern 1: Is it known that Xenopus Levis or zebrafish can also be used for regulatory testing of drug candidates? Or can the species only be used for basic research and screening? If possible, the topic should be addressed in the introduction or in the outlook.

We appreciate the reviewer’s feedback, and we have modified the text accordingly (lines 68 – 74).

Concern 2: Figure 1 corresponds in large parts to the figure from the cited paper. However, terms such as spasm and eyelid myoclonus are no longer listed. Are there reasons for this? Otherwise, all aspects of the original work should be taken up.

In the original publication, spasms and eyelid myoclonus are not in the basic operational classification and only listed under the expanded figure. Since these symptoms are subclassified under another category (e.g. eyelid myoclonia is listed under absence seizures) we opted to stick with only major classifications.

Concern 3: Mammals (e.g. mice) can also be kept in large numbers. The statement should be weakened. Rather, larvae and other developmental stages can of course be housed in a smaller area. For adult frogs, on the other hand, there are also limits to stocking densities. 

We have modified the text accordingly.

Concern 4: Can it be stated since when clawed frogs have been used in epilepsy research. References?

We have added a paragraph on the history of use of Xenopus in epilepsy research (lines 76-84). 

Concern 5: Are there also limitations in the methods and clear advantages in the implementation for aquatic species- The individual aspects (advantages and disadvantages in dosing, sampling) should be clearly presented separately. 

We thank the reviewer for this comment as we think new separate sections has added clarity to the manuscript (lines 100-168).

Concern 6: Figure2: Pre- and postsynapse could still be labelled. Also the colour of the neurotransmitters (excitatory/inhibitory). A small legend to the illustration can help understanding.

We have added labels to the figure to help with clarity.

Minor points: 

  1. Abstract: Add “clawed frog” (Xenopus) or even African clawed frog

The text has been corrected.

  1. Keyword: Add seizures, drug screening

The text has been corrected.

Reviewer 2 Report

The review by Williams et al, discusses the use of Zebrafish and Xenopus for seizure research as induction models, including using chemicals and genetic methods. They highlight the questions generated from several studies and further applications of these models to promote drug discovery for the treatment of epilepsy. The method of the search applied for the review of the literature is not well described, however the results are presented clearly and this made the reading of the manuscript very easy, even though there are missing information. I think that the manuscript gives new perspectives and arise new questions on the use of these models for epilepsy research, therefore deserved to be published on International Journal of Molecular Sciences after major revisions. I have some concerns listed below:

Major revisions:

1.      I would suggest to implement the paper adding information on the literature search carried out, by following the PRISMA guidelines as suggested by the author’s guidelines.

2.      In the results there is a literature gap, indeed I would suggest to repeat a literature search and add more information, in particular I would suggest to expand the paragraph 2.1, where several studies on the novel methods for seizures analysis are missing such as:

- Cozzolino O, et al. Evolution of Epileptiform Activity in Zebrafish by Statistical-Based Integration of Electrophysiology and 2-Photon Ca2+ Imaging.

- Hadjiabadi D, et al. Maximally selective single-cell target for circuit control in epilepsy models. Neuron.

- Lee Y, et al. An EEG system to detect brain signals from multiple adult zebrafish. Biosens Bioelectron. 2020 Sep 15;164:112315.

3. I suggest to explain better the limits of the most used methods (behavior and EEG recordings) for seizures analysis in both models

4. The paragraph 3.2 needs to be expanded, indeed there is no mention on several CRISPR/cas9 and morpholino models and the related drug screenings, in this case I would suggest also to add a table summarizing the screenings on different animal models, that could help the reader.

5. In the conclusion the authors should underline better the open questions on the use of the aquatic models for epilepsy research, explaining the potential pitfalls and giving some potential indications to overcome them.

Author Response

Reviewer 2

Concern 1: I would suggest to implement the paper adding information on the literature search carried out, by following the PRISMA guidelines as suggested by the author’s guidelines.

We appreciate the reviewer’s comment; however the editorial staff pitch back the first version of our manuscript as they required a percentage of the references to be <5 yrs old. Therefore, our literature search was less systematic than originally designed or required for a meta-analysis. Therefore, we chose not to include such a chart.

Concern 2: In the results there is a literature gap, indeed I would suggest to repeat a literature search and add more information, in particular I would suggest to expand the paragraph 2.1, where several studies on the novel methods for seizures analysis are missing such as:

- Cozzolino O, et al. Evolution of Epileptiform Activity in Zebrafish by Statistical-Based Integration of Electrophysiology and 2-Photon Ca2+ Imaging.

- Hadjiabadi D, et al. Maximally selective single-cell target for circuit control in epilepsy models. Neuron.

- Lee Y, et al. An EEG system to detect brain signals from multiple adult zebrafish. Biosens Bioelectron. 2020 Sep 15;164:112315.

We apologize for the oversight and have added the suggested references.

Concern 3:  The paragraph 3.2 needs to be expanded, indeed there is no mention on several CRISPR/cas9 and morpholino models and the related drug screenings, in this case I would suggest also to add a table summarizing the screenings on different animal models, that could help the reader.

We have expanded the text (lines 289-305).

Concern 4:  In the conclusion the authors should underline better the open questions on the use of the aquatic models for epilepsy research, explaining the potential pitfalls and giving some potential indications to overcome them.

We agree and have expanded the text (lines 359-373).

Reviewer 3 Report

Evaluator’s report on review submitted to International Journal of Molecular Sciences

Aquatic freshwater vertebrate models of epilepsy pathology: past discoveries and future directions for therapeutic discovery

By Rachel E. Williams and Karen Mruk

May 2022

The paper addresses the value of aquatic freshwater vertebrate models of epilepsy to the discovery of new therapeutic approaches to human epilepsy. Several review papers on this, or closely related, subjects have been published in the last two years and this review does not appear to bring much novelty or enhanced discussion to what has previously been advanced. Examples:

- Seizing the moment: Zebrafish epilepsy models. Gawel K, Langlois M, Martins T, van der Ent W, Tiraboschi E, Jacmin M, Crawford AD, Esguerra CV. Neurosci Biobehav Rev. 2020 Sep;116:1-20. doi: 10.1016/j.neubiorev.2020.06.010. PMID: 32544542

- Past, present and future of zebrafish in epilepsy research. Yaksi E, Jamali A, Diaz Verdugo C, Jurisch-Yaksi N. FEBS J. 2021 Dec;288(24):7243-7255. doi: 10.1111/febs.15694. PMID: 33394550

- Age Bias in Zebrafish Models of Epilepsy: What Can We Learn From Old Fish? Cho SJ, Park E, Baker A, Reid AY. Front Cell Dev Biol. 2020 Sep 10;8:573303. doi: 10.3389/fcell.2020.573303. PMID: 33015065

- Post-Traumatic Epilepsy in Zebrafish Is Drug-Resistant and Impairs Cognitive Function. Cho SJ, Park E, Baker A, Reid AY. J Neurotrauma. 2021 Nov 15;38(22):3174-3183. doi: 10.1089/neu.2021.0156. PMID: 34409844

The paper makes an overview of the advantages of studying epilepsy in these models, but the article addresses several of the topics very lightly and it is not very accurate in some of its statements/descriptions that require several improvements. It appears that words are missing in several parts of the text and the manuscript should be carefully checked for these errors.

As such, I do not recommend this paper for publication at this point.

The paper should be better constructed to elaborate in more detail the mechanisms and usefulness of these models, and put in context the most recent advances in the field.

On page x, line 44, the authors mention ‘seizures in newborns’ and in the next sentence ‘In contrast, neonatal-onset seizures…’. It is not clear at all for the readers what is the contrast between the two.

I believe it should be stated something like: ‘Seizures in newborns are usually due to neonatal brain injury (hypoxia, ischemia, stroke, or intracranial hemorrhage) being classified as acute symptomatic seizures and [14]. Less frequently, neonatal‐onset seizures can be due to genetic mutations [15‐19], brain malformations [20,21], or metabolic disorders [22, 23].’

On page 3, line 74 the authors claim that ‘…children with epilepsy have abnormal myelin [63, 64],’

I think the correct statement is abnormal myelination patterns, since I believe there is no mutations in myelin described.

On page 3, line 106 authors refer ‘The ease of genetic manipulation in zebrafish and Xenopus affords them a major advantage over mammalian models as genetically induced seizures better simulate the clinical situation than other experimental models. For example, genetic seizure disorders are highly variable and prone to drug resistance [88].’

R: This statement is in concept incorrect. Modelling using genetically modified vs chemically challenged or kindled animals is more related to the type of epilepsy one whishes to model than with the fact that genetically induced seizures reproduce more closely the clinical situation. In fact, there are a number of cases of epilepsy in humans that are secondary to brain lesions, seizures, AVC. I do not believe at all these genetically induced seizures reflect better these types of acquired epileptogenesis.

On page 4, line 132, the sentence seems incomplete ‘…Therefore, most chemically induced models of seizures in aquatic vertebrates use compounds that will disrupt the inhibitory and excitatory balance in the brain of the animal (Figure 2).’

On page 4, line 136, ‘…non‐PTZ chemical induction methods have been exclusively used in zebrafish.’ Is this as compared to frogs? Because they have been extensively used in rodents. Authors should be more precise in their statements. The underlined verb (or any other) appears to be also missing in this sentence.

Figure 2 is not very useful as it only illustrates the mechanism of action of some of the represented compounds, making it difficult to apprehend what the others do. Furthermore, it is a bit overcrowded with compounds and oversimplifies the mechanism of drugs (e.g., GABAA receptors are not exclusively postsynaptic,…).

The discussion on the applicability of the different drugs for tank administration in fish and frogs appears useful.

On Page 5, line 157 it is stated ‘Picrotoxin is also a GABAA antagonist but unlike bicuculline, is a non‐completive inhibitor.’ I think the authors mean it is a non-competitive inhibitor.

Minor points:

1. Numbering in the references section is doubled.

2. Reference number 1 does not seem to be correctly cited. The weblink is also invalid.

3. On page 3 line 70, ‘…and transparency describe above…’ should be ‘…and transparency described above…’ 

4. On page 3, line 92, ‘…pharmacokinetic challenges such indirect measurements…’: should it be ‘…pharmacokinetic challenges such as indirect measurements…’?

Author Response

Reviewer 3

Concern 1: . Several review papers on this, or closely related, subjects have been published in the last two years and this review does not appear to bring much novelty or enhanced discussion to what has previously been advanced.

We agree with Reviewer #2 that many reviews exist on the utility of zebrafish as an epilepsy model. However, to our knowledge, there are no reviews about Xenopus models. Furthermore, the zebrafish reviews do not mention the use of other freshwater vertebrates such as Medaka. Therefore, we believe we are adding to the traditional zebrafish discussion. We have added more information about Xenopus to create a more balance review.

Concern 2: On page x, line 44, the authors mention ‘seizures in newborns’ and in the next sentence ‘In contrast, neonatal-onset seizures…’. It is not clear at all for the readers what is the contrast between the two.

We have modified the text to differentiate between a seizure and epilepsy.

Concern 3: On page 3, line 74 the authors claim that ‘…children with epilepsy have abnormal myelin [63, 64],’

We have modified the text.

Concern 4: Modelling using genetically modified vs chemically challenged or kindled animals is more related to the type of epilepsy one whishes to model than with the fact that genetically induced seizures reproduce more closely the clinical situation. In fact, there are a number of cases of epilepsy in humans that are secondary to brain lesions, seizures, AVC. I do not believe at all these genetically induced seizures reflect better these types of acquired epileptogenesis.

We agree with the reviewer that genetic models do not better reflect acquired epileptogenesis. Our aim was to highlight that genetic models best model drug-resistant seizures that are seen in the clinic. We have modified the text to make this more clear (lines 127-143).   

Concern 5: Figure 2 is not very useful as it only illustrates the mechanism of action of some of the represented compounds, making it difficult to apprehend what the others do. Furthermore, it is a bit overcrowded with compounds and oversimplifies the mechanism of drugs (e.g., GABAA receptors are not exclusively postsynaptic,…).

We have reworked Figure 2 to include better labels and spaces between the molecules. All of the molecules we discuss are pictured.

Minor points: 

  1. Numbering in the references section is doubled.

The doubling appears to be due to EndNote annotation and the MDPI template. We will work with copy editors to correct the issue.

  1. Reference number 1 does not seem to be correctly cited. The weblink is also invalid.

The weblink was correct; however, when EndNote added the citation the https portion was not hyperlinked causing an error. 

  1. On page 3 line 70, ‘…and transparency describe above…” should be ‘…and transparency described above…’

The text has been corrected.

  1. On page 3, line 92, ‘…pharmacokinetic challenges such indirect measurements…’ should it be pharmacokinetic challenges such as indirect measurements…’

The text has been corrected.

Round 2

Reviewer 2 Report

The revised version of the manuscript has been improved as suggested, therefore can be accepted for publication.

Author Response

We thank the reviewer for their time evaluating the manuscript and appreciate the positive response.

Reviewer 3 Report

Evaluator’s report on review submitted to International Journal of Molecular Sciences V2

Aquatic freshwater vertebrate models of epilepsy pathology: past discoveries and future directions for therapeutic discovery

By Rachel E. Williams and Karen Mruk

July 29th, 2022

The paper addresses the value of aquatic freshwater vertebrate models of epilepsy to the discovery of new therapeutic approaches to human epilepsy. In this revised version the quality of the paper has greatly improved. However, there are a few points that authors still need to address.

1.       On page 7, Line 261, when mentioning ‘…the efficacy of current anti‐seizure compounds was tested in rat slices with 4‐AP‐induced seizures’, the brain area of the rat slices should be mentioned for clarity. Also, replace tested by performed as you already mentioned a test in the same sentence.

2.       On page 7, Line 271, when mentioning ‘…anti‐seizure compounds from these screens in zebrafish larvae are similar to those identified in rodent studies’ If I understood correctly, I think what the authors mean is that those anti-seizure compounds are consistent (and not similar compounds) to those in other studies. If so, text should be corrected accordingly.

3.       On page 9, Line 336, the meaning of the sentence is not entirely clear for someone that has not read the referenced paper. Were the authors studying seizures induced by locomotor activity? Or locomotor activity during seizures? Please clarify.

4.       The beginning of the conclusion section is a bit confusing. The authors review several seizure/epilepsy models that are already used in aquatic models and their relevance for studying epilepsy. Yet, they begin their conclusion saying there are not alternatives to PTZ seizure induction (I think they mean in what regards drug-resistant epilepsies specifically). Nevertheless, this section should start with a more general comment on the main topics reviewed. The second sentence is also not very clear. I think what the authors mean is that there is no complete superimposition on the anti-seizure efficacy (not activity) of all clinically used antiepileptic drugs in each of (and not a single) the seizure induction models currently used in aquatic seizure models. I think the whole first paragraph is still not clear and needs to be readdressed, namely in clarifying that research in aquatic seizure models may help uncover new epilepsy precipitating drugs that may generate better models of drug resistant epilepsy.

Minor points:

Page 3, Lines 88-90 – Two consecutive sentences start with ‘In addition,…’

Page 3, Lines 104 – replace ‘…assess rate…’ by ‘…assess the rate…’

Page 4, Lines 140 – replace ‘A unique…’ by ‘An unique…’

Page 5, Lines 159 – replace ‘…a EEG…’ by ‘…an EEG…’

Page 6, Lines 205 – replace ‘…seizure behaviors…’ by ‘…seizures…’

Page 6, Lines 211 – replace ‘…seizure …’ by ‘…seizures…’

Page 7, Lines 274 – replace ‘…medication …’ by ‘…treating…’

Page 8, Lines 282 – replace ‘…seizure …’ by ‘…seizures…’

Page 8, Lines 294 – replace ‘…have …’ by ‘…has…’

Author Response

Reviewer #3

The paper addresses the value of aquatic freshwater vertebrate models of epilepsy to the discovery of new therapeutic approaches to human epilepsy. In this revised version the quality of the paper has greatly improved. However, there are a few points that authors still need to address.

 We thank the reviewer for their positive comments.

  1. On page 7, Line 261, when mentioning ‘…the efficacy of current anti‐seizure compounds was tested in rat slices with 4‐AP‐induced seizures’, the brain area of the rat slices should be mentioned for clarity. Also, replace tested by performed as you already mentioned a test in the same sentence.

      We have edited the text.

  1. On page 7, Line 271, when mentioning ‘…anti‐seizure compounds from these screens in zebrafish larvae are similar to those identified in rodent studies’ If I understood correctly, I think what the authors mean is that those anti-seizure compounds are consistent (and not similar compounds) to those in other studies. If so, text should be corrected accordingly.

      We agree this sentence was not clear and have modified it.

  1. On page 9, Line 336, the meaning of the sentence is not entirely clear for someone that has not read the referenced paper. Were the authors studying seizures induced by locomotor activity? Or locomotor activity during seizures? Please clarify.

      We added a description of the experimental parameters.

  1. The beginning of the conclusion section is a bit confusing. The authors review several seizure/epilepsy models that are already used in aquatic models and their relevance for studying epilepsy. Yet, they begin their conclusion saying there are not alternatives to PTZ seizure induction (I think they mean in what regards drug-resistant epilepsies specifically). Nevertheless, this section should start with a more general comment on the main topics reviewed. The second sentence is also not very clear. I think what the authors mean is that there is no complete superimposition on the anti-seizure efficacy (not activity) of all clinically used antiepileptic drugs in each of (and not a single) the seizure induction models currently used in aquatic seizure models. I think the whole first paragraph is still not clear and needs to be readdressed, namely in clarifying that research in aquatic seizure models may help uncover new epilepsy precipitating drugs that may generate better models of drug resistant epilepsy.

 We agree that the first paragraph was still unclear. We have rewritten the conclusion section.

Minor points:

Page 3, Lines 88-90 – Two consecutive sentences start with ‘In addition,…’

We deleted the second in addition.

Page 3, Lines 104 – replace ‘…assess rate…’ by ‘…assess the rate…’

We fixed the text.

Page 4, Lines 140 – replace ‘A unique…’ by ‘An unique…’

The beginning of unique has a consonant sound and thus should be A not An

Page 5, Lines 159 – replace ‘…a EEG…’ by ‘…an EEG…’

We fixed the text.

Page 6, Lines 205 – replace ‘…seizure behaviors…’ by ‘…seizures…’

We changed the text.

Page 6, Lines 211 – replace ‘…seizure …’ by ‘…seizures…’

We changed the text.

Page 7, Lines 274 – replace ‘…medication …’ by ‘…treating…’

We fixed the text.

Page 8, Lines 282 – replace ‘…seizure …’ by ‘…seizures…’

We fixed the text.

Page 8, Lines 294 – replace ‘…have …’ by ‘…has…’

We fixed the text.